# Energy Electronegativity and Chemical Bonding

**DOI:** 10.3390/molecules27238215

**Published:** 2022-11-25

**Authors:** Stepan S. Batsanov

**Affiliations:** National Research Institute for Physical-Technical Measurements, 141570 Mendeleevo, Russia; batsanov@mail.ru

**Keywords:** electronegativity, bond ionicity, solid state, nanophases, oxidation reaction

## Abstract

Historical development of the concept of electronegativity (EN) and its significance and prospects for physical and structural chemistry are discussed. The current cutting-edge results are reviewed: new methods of determining the ENs of atoms in solid metals and of bond polarities and effective atomic charges in molecules and crystals. The ENs of nanosized elements are calculated for the first time, enabling us to understand their unusual reactivity, particularly the fixation of N_2_ by nanodiamond. Bond polarities in fluorides are also determined for the first time, taking into account the peculiarities of the fluorine atom’s electronic structure and its electron affinity.

## 1. Introduction

Of some 10^8^ substances known today, only a few hundred (i.e., elementary solids and molecules) are homo-nuclear, all the rest containing polar bonds. To describe the polarity, Avogadro (1809) and Berzelius (1811) listed elements according their ‘negative’ or ‘positive’ electrochemical characters. The development of this approach led to a conclusion that the bond polarity is caused by a shift of valence electrons towards the atom that attracts them more strongly, i.e., that with the higher EN [1,2]. Thus, by comparing the ENs of the bonded atoms, we find which of them has a negative charge and how large it is. The full history of the EN concept is described in detail by Jensen [3,4] and Sproul [5]. Remarkably, the popularity of this concept has not decreased in 200 years–on the contrary, in recent years it has begun to be used to solve new problems of materials science, structural chemistry and high-pressure physics and chemistry.

Fajans [6,7] created the theory of polarization to study chemical bonding from the ionic viewpoint, but this approach cannot quantify the chemical bond since the radii (and hence the energy of polarization) of ‘pure’ ions cannot be experimentally found [8], whereas the properties of covalent bonds can be measured accurately, yielding the basis for a quantitative scale of ENs.

The modern age of the EN concept began in 1932 when Pauling [9] found that
χ_M_ − χ_X_ = *a* [*E*_MX_ − ½ (*E*_MM_ + *E*_XX_)]^½^ = *a* Δ*E*_MX_^½^(1)
where χ is the EN, *E* is the bond energy, and *a* = 0.208 or 0.102 if the energy is expressed in kcal/mol or kJ/mol, respectively (Pauling then expressed *E* in eV, for which *a* = 1—this is the now well-forgotten origin of his unit of EN!).Thus, Pauling showed that the thermal effect of a reaction is a function of charges on the atoms. This work initiated numerous studies deriving EN from, or relating it to, a wide variety of physical properties of materials. Here, we will limit ourselves only to the energy-based system of ENs and its relation to the effective charges of atoms in molecules and crystals. Section 2.1, Section 2.2 and Section 2.3 provide a concise review of the development of the EN concept from 1932 to the present day, as applied to free atoms, atoms in molecules and in solids. Section 2.4 contains the author’s new research on applying the EN concept to explain peculiar properties of solid nanoparticles, rationalizing their high reactivity and deriving (for the first time) a specific EN system for nanomaterials. Section 3.1, Section 3.2 and Section 3.3 review the applications of EN to the problem of effective atomic charges and bond polarities in molecules and solids (Section 3.3). Section 3.2 also contains new calculations for compounds and bonds containing fluorine, a stumbling-block for the EN approach due to its peculiar electronic configuration. Section 3.4 and Section 3.5 review the application of EN to coordination compounds of transition metals (including some counter-intuitive, but experimentally validated, predictions) and to the high-pressure behavior of substances. The concluding Section 4 briefly surveys various critiques—scientific, philosophical and ideological—of the EN concept.

## 2. Electronegativity

### 2.1. Electronegativity of Free Atoms

Mulliken was the first to express the ENs of atoms through their ionization energies [10,11]:χ = ½ (*I_v_* + *A_v_*)(2)
where *I_v_* is the first ionization potential and *A_v_* is the electron affinity for the valence states of the atoms. Between Mulliken’s and Pauling’s values of the EN, there is a straightforward relation: χ_P_ = 0.36 χ_M_. Unfortunately, there is serious ambiguity in specifying the valence state of some atoms; e.g., for N and O one has to choose from 7 and 8 possible states, respectively, with ionization energies and electron affinities covering ranges of 10 or more eV [12]. Pearson [13] used the ground-state ionization energy and electron affinity of the atom (*I_o_* and *A_o_*) to calculate the EN by Equation (2). Although his ENs generally show the expected trends in the Periodic Table, there are some strikingly unrealistic values. These violations disappear if the *average* (rather than the first) ionization potentials for the ground state are used. Obviously, if all outer valence electrons of an atom are involved in the chemical bonding, they become equivalent and their energy must be characterized by the average potential:(3)$$\bar{I}=\frac{1}{v}\sum_{i=1}^{v}I_{i}$$

Converting to the dimensionality of Pauling’s EN (square root of energy), we obtain [14]:(4)$$\chi =c{\left(\bar{I}\right)}^{1/2}$$
which gives good agreement with the standard system of ENs. The $\bar{I}$ were used for calculating ENs also in [15,16,17]. Table 1 lists the ENs from [14], improved in some cases by the use of up-to-date values of ionization potentials for polyvalent elements [18]. Mulliken’s approach was developed further by many authors (see reviews in [19,20]).

### 2.2. Electronegativity of Atoms in Molecules

Equation (1), or its modifications, were applied by many authors to calculate the ENs of elements for various valence states, using ever more extensive and precise experimental data (see a survey in [16]). However, it is really valid for low-polarity molecules only, for the bond energy depends on many factors besides the polarity, particularly on the bond distance, atomic valences and hybridization states. An elegant solution to these difficulties was found by Tantardini and Oganov [21] who proposed the equation:(5)$$E_{\mathrm{MX}}\;=E_{\mathrm{MX}}^{\mathrm{cov}}\;(1+\Delta \chi_{\mathrm{MX}}^{2}))$$
where $E_{\mathrm{MX}}^{\mathrm{cov}}$ is the covalent component of the bond energy, and the EN difference is the measure of the relative (rather than absolute, as in Pauling’s approach) ionic contribution. Equation (5) allows us to obtain correct trends of EN in the Periodic Table and to eliminate some anomalies in both the absolute and the relative values of the ENs yielded by Equation (1), such as those of the alkali metals (Li 1.85, Na 1.96, K 1.99, Rb 2.00, Cs 1.93). Tantardini–Oganov’s ENs give a reasonable prediction of bond polarity and improve the description of the thermal effects of chemical reactions. However, these ENs differ so much from the generally accepted values that it is difficult to use them in previously found functional dependencies.

Therefore, Batsanov [22] improved Equation (1) purely empirically while preserving its philosophy. For MX*_n_* type compounds, this was achieved by replacing the constant *a* = 0.102 with a variable parameter *c =* 0.1 + 0.015(*n** + *d* – *v*) where *n** is the ratio of the principal quantum numbers of the metal and nonmetal, *d* is the bond length (in Å), and *v* is the valence of M. This correction takes into account that bond polarity is enhanced by an increase in bond length (reducing the overlap of the valence orbitals) and of *n** but lowered when the valence increases. Table 2 lists the average ENs calculated by this procedure using, for non-metals, χ = 3.9 (F), 3.1 (Cl), 2.9 (Br) and 2.6 (I).

### 2.3. Electronegativity of Atoms in Solids

Most EN systems are based on the properties of molecules which retain their structures in all aggregate states, but in inorganic solids with covalent, metallic or ionic bonds the situation drastically differs owing to the large cooperative effect. Despite this obvious difference, conventional ‘molecular’ ENs are still routinely applied to ionic crystals, metallic alloys or coordination compounds. Thus, the authors of [23,24,25,26] derived the ENs of elements from the physical properties of solids, while using the ‘molecular’ ENs of C, N, O as reference points, although Duffy [27,28] had shown that the ENs of halogen, oxygen and chalcogen atoms also differ somewhat in the solid state.

The idea of specific solid-state ENs for crystals was developed by Batsanov [29]. The ‘chemical scale’ of Pettifor [30,31] showed clear separation of different structures of binary compounds (MX) in two-dimensional maps in the coordinates χ_M_ and χ_X_, and although he used Pauling’s ENs for Be through F, curiously enough, for most elements his values were close to the solid-state ENs of Batsanov [29]. The latter also adapted Equation (2) for the solid state by replacing the ionization potentials of free atoms with the work functions (*Φ*) as their analogs for the solid state [32] and the electron affinities of isolated atoms with those of atoms in solids (*A_s_*), measured by secondary ion mass spectrometry [33]:χ*_s_* = ½ (*Φ* + *A_s_*)(6)

Usually, Mulliken’s ENs are converted to Pauling’s scale (χ*_o_*) with the help of a constant factor [10], but more correctly, this conversion should be element-specific rather than uniform, by using Equation (7):
(7)$$\frac{\chi_{s}}{\chi_{o}}=\frac{\Phi +A_{s}}{I_{o}+A_{o}}$$
where χ*_o_* is the thermochemical EN in Pauling’s scale, referring to the lowest ionization state. The results of calculations of $\chi_{s}$ by Equations (6) and (7) are listed in Table 3.

The thermochemical ENs of atoms in crystalline compounds MX*_n_* (Table 4) can be calculated by Equation (1), replacing Δ*E*_MX_ with the enthalpy of the formation of compounds in the solid state (*Q_s_*):Δχ_s_ = *a* (*Q_s_*/*n*) ^½^
(8)
where *a* = 0.1 + *cn** and *c* = 0.01 for the M^*n*+^ cations with a noble-gas electron configuration, or 0.02 otherwise [32].

### 2.4. Electronegativity of Nanosized Elements

Experimental data and all theoretical models show that the cohesive energy of nanoparticles decreases linearly with their size (see the review in [34]). From the structural point of view, a decrease in the particle size reduces the average coordination number in it, as the fraction of surface atoms (with lower coordination numbers) relatively increases. Ultimately, in a 13-atom metal cluster, all atoms but one lie on the surface, and in an 8-atom fragment of a diamond-like structure—all atoms. The cohesive (atomization) energy (*E_p_*) of a particle of arbitrary size is related to that of the bulk solid (*E_b_*) as
(9)$$E_{p}=E_{b}\;\frac{{\bar{N}}_{p}}{N_{b}}$$
where $\bar{N}$*_p_* is the average coordination number of atoms in the particle, and *N_b_* is that in the bulk [35]. Methods of calculating the coordination numbers of atoms on the surface of cubic crystals are described in [36,37], and the results of these calculations are given in Table 5 for 5 nm-sized particles.

Of course, $\bar{N}$*_p_* is only the average: the surface atoms are different from those in the interior of a particle, and while the ‘physical’ averaging does occur to some degree, its extent and character depends on the nature of the chemical bond in the particle. It is naïve indeed to imagine the surface bristling with broken, dangling bonds—such a particle will be too reactive to exist. The surface can be terminated by alien atoms or functional groups—e.g., nanodiamond (ND) particles are terminated by C-H bonds or by -OH, -COOH, etc., groups [38]—and thus become chemically different from the bulk. If such termination is absent or stripped away, the outer layer of ND reconstructs into a shell of sp^2^ carbon atoms, variously described as graphitic, fullerene- or onion-like [39], with an underlying thicker (up to 5 Å) layer of amorphous sp^3^ carbon [40]. In metals, the relaxation may take the form of stronger (and shorter) bonding between the first and the second outermost layers of atoms, compensated by longer distances between the second and the third, with such alternating distortions perpetrated for several layers inward [41].

A decrease in the cohesive energies in Cu, Au, Pt, C, Si, or Ge particles of such sizes by an average of ca. 60 kJ/mol will enhance their reactivity. Indeed, we found that complete evaporation of aqueous transparent Si or Ge colloids yields solid residues containing SiO_2_ or GeO_2_ microcrystals of the quartz type with refractive indices *n*_e_ = 1.553, *n*_o_ = 1.544 and *n*_e_ =1.735, *n*_o_ = 1.695, respectively. In the case of water colloids of carbon, we observed a mass loss of carbon equal to 0.02 wt % due to the formation of CO_2_, which either escapes or is consumed in the growth of fibers often formed from such colloids [42]. Thus, the nanosized grains of E = C, Si and Ge replace hydrogen in water *under ambient conditions*.
E + 2H_2_O = EO_2_ + 2H_2_

In fact, similar conclusions for Cu and Pt have been occasionally made earlier. Thus, it was shown [43] by DFT calculations that the cohesive energy of Cu atoms decreases linearly with their coordination numbers in particles and, therefore, copper nanoparticles have a lower equilibrium potential for dissolution in water. A full thermodynamic analysis [44,45] of size effect on the standard potential defining Pt/PtO equilibrium, has shown that under certain conditions PtO may form in water solutions.

The size effect in the cohesive energy of carbon was studied by measuring the oxidation heat of polycrystalline synthetic diamond (PD) with *D* = 146 μm, and carbon nanotubes (CNT) with *D* = 50–70 μm.

In addition, detonation nanodiamond (DND) with *D* = 5 nm [37] was also studied. The results, shown in Figure 1, indicate that the smaller the particle size is, the higher the combustion heat (viz. Δ*H*_ox_ = 29.7, 34.4 and 38.6 kJ/g for PD, CNT and DND, respectively), hence the cohesive energy is lower. Although CNT is sp^2^ carbon, while PD is sp^3^ carbon and DND mostly so, this difference is far too small to account for the observed effect: note that the enthalpy of the diamond to graphite phase transition is ca. 0.2 J/g.

From Table 5, it is evident that the difference between *E_b_* and *E_p_* for diamond (58 kJ/mol on average) is big enough even to change the sign of the thermal effect of a chemical reaction, as we observed in the case of fixation of the atmospheric nitrogen by the 5-nm sized detonation nanodiamond [42,46]. Over time, diluted aqueous colloids (suspensions) of DND gave rise to cottonwool-like fibers with the nitrogen content and nitrogen to carbon ratio higher than in the DND used, the only possible source of the excess nitrogen being the atmospheric, or specially injected, N_2_. One can envisage a three-stage process:2C + N_2_ = C_2_N_2_
C_2_N_2_ + 4H_2_O = 2CO_2_ + 2NH_3_ + H_2_
C_2_N_2_ + 3O_2_ = 2CO_2_ + 2NO

The second and especially the third reaction are indeed consistent with the gaseous product detected by mass-spectroscopy [46]; they are also both exothermic, with Δ*H* = −47.2 and −297.5 kJ mol, respectively. The first stage, however, is highly unfavorable thermodynamically (Δ*H* = 308.9 kJ mol) under standard conditions and would require temperatures above 1400 °C to proceed. However, this refers to a bulk diamond with the atomization energy *E_a_* = 716.7 kJ mol; the reaction would be exothermic for atomic carbon (Δ*H* = −1124.5 kJ mol) or C_2_ molecules (Δ*H* = −506.5 kJ mol). A 22% decrease in the *E_a_*(C) would make this reaction exothermic. Admittedly, C_2_N_2_ was not detected, which can be due to its rapid hydrolysis or oxidation.

Obviously, as the particle size decreases further, $\bar{N}$*_p_* will approach the coordination number of the surface atoms, *N_s_*, and in this case, the surface energy, *E_s_*, will be
(10)$$E_{s}=E_{b}\;\frac{N_{s}}{N_{b}}$$

Using the *E_s_***/***I_o_* ratio for each element as a conversion factor, one can calculate *I_s_* for the extremely small nanoparticles, and then with the help of Equation (4) find the ENs of nanophases of elements (Table 6).

## 3. Effective Charges of Atoms

### 3.1. Bond Polarity in Molecules

Effective charges on atoms can be neither directly measured nor theoretically defined in a unique manner, for the obvious reason that the electron shells of atoms in molecules or solids overlap, but nevertheless are very useful in chemistry. They can be deduced from observable parameters using a variety of models, from purely empirical and mechanistic to grounded in quantum chemistry, such as QTAIM [47]. Meister and Schwarz [48] surveyed 26 such models and found them to have the same underlying physics. It is the electron density (ED) distribution that determines all chemical and physical phenomena and may be measured, for instance, by high-resolution XRD experiments [49]. Thus, the task is reduced to finding the ED in a certain region of an atom and comparing it with that in a free atom.

Currently, the most popular is the evaluation of the effective charges of atoms in molecules from dipole moments, *μ*
*μ* = *qd*(11)
where *q* is the atomic charge, and the dipole length *d* can be identified with the bond length. Then, the *μ*/*d* ratio should characterize the polarity of the chemical bond, *p*. This ratio was used by Fajans to estimate the bond polarities in HCl and HBr molecules [50], and later Pauling [51] used it to determine the ionicity (polarity) in any chemical bond as
(12)$$i=1-e^{-X}$$
where *X* = *c*Δχ^2^ (initially *c* = 0.25 was suggested, and later *c* = 0.18 [52]). Table 7 lists the dipole moments of diatomic molecules (from [18], unless specified) and their polarity of bonds calculated as *p* = *μ*/4.8 *d*, with *μ* in Debye units (*D*) and *d* in Å.

However, Equation (11) is a rather crude simplification. Coulson [55] concluded that the *μ* of a bond consists of several terms, caused by the EN difference (*μ*_i_), the electron density difference (*μ*_ρ_), the hybridization of the atomic orbitals (*μ*_h_), and the effect of non-bonding electrons (*μ*_e_). Thus, the bond ionicity can be found only from *μ*_i_, e.g., when other components of *μ* are known theoretically or empirically. Moreover, the assumption that the *d* in Equation (11) is the distance between the nuclei of atoms is also only the first approximation. Owing to the flexibility of the electron clouds around each nucleus, the effective charge center is not at the nucleus, and the accurate dipole moments should be equal to *μ* = *qR*_c_, where *R*_c_ is the separation between these charges [56]. Authors of [57] indicated that atomic charges in a highly ionic molecule can be computed only if the molecular dipole moment, the polarizabilities of free cations and neutral atoms of the anions, are all known. In [58], a new approach to calculating the local dipole moments and charges of atoms based on Bader’s AIM theory has been presented. Further development of this approach is based on the study of the infrared intensities of fundamental bands in gaseous molecules, which contain a wealth of information about electronic structure and its changes on molecular vibration. The authors of [59,60,61,62,63,64] showed that from these experimental data, one can obtain values of the bond ionicity in molecules (Table 8), which are qualitatively consistent with the results of calculations by Equation (12).

### 3.2. Equalization of Electronegativity of Atoms in Molecules

In this context, the idea expressed by Pauling [65] that in stable compounds ‘each atom has only a small electric charge, approximating zero’ is very important. Since the positive charge increases the EN of a metal atom (M), but the negative charge of nonmetal (X) reduces it, the ENs of atoms in an MX molecule must converge. Sanderson [66,67] developed this idea and postulated the full equalization of the atomic ENs in the molecule. Using, as a reference, the bond ionicity in the NaF molecule (0.75), Sanderson was able to calculate the effective charges of atoms in other molecules.

In the work of [68], it has been found that the energy of ionization M → M*^N^*^+^, expressed as a fraction of removing all valence electrons, shows a simple parabolic dependence on the degree of ionization; the ‘fractional charge’ *i = n* /*N*:
(13)$$\bar{I}=\sum_{n}I/\sum_{N}I=i^{2}$$

This dependence is a continuous and smooth function, despite the fact that the ionization of each isolated atom is a discrete process. As has been pointed out [69], the energy functions *E*(*N*) for atoms-in-molecules are differentiable with respect to *N*, although *N* has only integer values for isolated atoms.

Accordingly, the electron affinity
(14)$$\bar{A}=1-i^{2}$$

The intersection point of the *I* (*i*) and *A* (*i*) curves means that at M + X, the contact of the ionization energy *E_I_* = *i*^2^*I* and the electron affinity *E_A_* = (1 − *i*^2^)*A* are equal (Figure 2), hence:(15)$$i={\left(\frac{A}{A+I}\right)}^{1/2}$$

The values of *i* for halides MX*_n_* can be calculated under the assumption that all outer electrons in a metal atom are equivalent, involved in bonding, and, upon hybridization, become averaged in their properties. As a result, the average ionization potential $\bar{I}$*_n_* of all valence electrons turns out to be much more informative than *I*_1_ [14,15]. Use of the average ionization potential ${\bar{I}}_{n}$ in Equation (15) allows us to calculate *i* in the MX*_n_* molecules with a different *n*, i.e., finding *i* for different valences of the metal atoms.

An important reservation concerns applying Equation (15) to fluorides. According to the correlation found between the electron affinity and atomic sizes or ionization potentials of heavier halogens, fluorine must have a higher *A* value than the observed one. The difference can be explained by the destabilizing effect (in F) of a high concentration of negative charge in a small volume [70,71,72]. In fact, other Period 2 elements show a similar trend: the electron affinities of O, N, C, B are lower than those of S, P, Si, Al, respectively. Bonding of a fluorine atom to any other increases the volume available for its electrons and, hence, relieves the inter-electron repulsion. Therefore, a fluorine atom in compounds is better characterized by the ‘extrapolated’ value of *A* = 4.49 eV [70]. Table 9 presents the values of the bond polarities (*i*_B_) in MX molecules, calculated by Equation (15) using this *A* (F) and the standard values of electron affinities of other halogens and ionization potentials of metals from [18].

In Table 9, Batsanov’s values of *i*_B_ are compared with those calculated by Equations (12) and (16), i.e., according to Pauling, *i*_P_, or Sanderson (*i*_S_):(16)$$i_{S}=\frac{S_{\mathrm{MX}}-S_{M}}{\Delta S}$$
where *S* is the EN in Sanderson’s scale, *S*_MX_ = $\sqrt{S_{M}S_{X}}$ and Δ*S* = *S*_M+_ − *S*_M_ [66]. As is seen from Table 9, there is only a qualitative agreement between these methods, but both the Pauling’s and the Sandersen’s equations depend on the accepted reference points, while Equation (15) uses only the experimentally established dependence of the ionization potentials on the charges of atoms, and the values of these characteristics themselves are measured with great accuracy. Generally speaking, there are 26 different methods (!) for estimating the effective charges of atoms in compounds calculated by various models of quantum chemistry [73]; however, in this review, we limit these to empirical approaches only using the EN concept.

### 3.3. Effective Charges of Atoms in Solid Compounds

The first experimental estimates of the effective charges of atoms in crystalline compounds were based on the work of Born [74], who suggested that in inorganic crystals, the electric field acting on each ion is equal to the applied external field related to the dielectric permittivity, *ε*, and the refractive index, *n*, with the frequency of the transverse lattice vibrations (*ω*_t_):(17)$$\varepsilon =n^{2}+(4\pi e^{2}/{\omega_{t}}^{2}\bar{m}V)$$
where $\bar{m}$ is the reduced mass of the vibrating particle and *ω*_t_ is a frequency of the transverse oscillations.

A comparison of the experimentally measured and calculated values of *ε* showed that, usually, the computed *ε* are less than the experimental ones. Szigeti [75,76,77,78], taking into account that the field strength in the dielectric is less than the external field due to the polarization of the specimen, deduced the equation:(18)$$\varepsilon =n^{2}+\frac{4\pi e^{2}}{\omega_{o}^{2}\bar{m}V}\frac{{\left(n^{2}+2\right)}^{2}}{9}$$

The values of *ε* calculated by Equation (18) are always higher than those measured experimentally. This led him to the idea to replace the formal charge of the ion by the real effective charge *e**. Thus, Szigeti’s formula allows us to calculate the *e** in crystals, using the experimental values of *n* and *ε*, which are higher than those in molecules owing to the excess of coordination numbers over the valence of atoms in crystals.

For the calculation of *i* in the MX crystals in Equation (15) it is necessary to replace *I*_*o*_ with the work function, *Φ*, which is the ionization potential for a solid, and then
(19)$$i_{s}={\left(\frac{A}{A+\Phi }\right)}^{1/2}$$

Table 10 shows that our values of the *i* of atoms calculated by Equation (19) are close to the experimental data of Szigeti’s *e** [79].

The greater effective charges in halides CuX and AgX are caused by forming the additional bonds Cu, Ag →X of (*n* − 1)*d*-electrons in these metals, and Tl→X of *np*-electrons with vacant *nd*-orbitals in halogens [80,81].

### 3.4. Coordination Charges of Atoms in Chemical Compounds

Above, the ‘effective charge ’notion is interpreted as a synonym of ‘bond ionicity’ (in fractions of an electron). In reality, the term ‘charge’ means either an excess or a deficit of electrons in the given region of the atom, as compared to its free state. Therefore, it is important to determine the region of the atomic space to which the charge being determined relates. Batsanov [82] had proposed to divide the effective charges determined by different methods into two groups—proper (*q*_o_) and coordination (*q*_c_) charges. If the first term characterizes the situation inside the closed orbitals of an atom, then the second type is a result of the electron interaction of the given atom with its neighbors in the first coordination sphere.

This *q*_o_ defines the Coulomb energy, is responsible for the IR absorption, causes the atomic polarization bands, and affects the binding energy of the internal electrons in the atom. However, what matters for redox reactions is the electron density in the interatomic space, i.e., the *q*_c_. Therefore, Suchet [83], instead of using the terms ‘proper’ and ‘coordination’ as they refer to the charges of atoms, used the terms ‘physical’ and ‘chemical’.

The notion ‘charge on the atom’ and different methods of its determination have been discussed by Catlow and Stoneham [84]. From the above it follows that
*q*_o_ = *vi*, *q*_c_ = *v* ± (1 − *i*) *N*_c_(20)
where *v* is a valence and *N*_c_ is the coordination number of the atom in the crystal structure. According to Equation (20), in crystalline compounds with low bond polarity, *q*_c_ can become negative if *N*_c_ > *Z* (see details in [18]). Interestingly, in 1948, Pauling [65] calculated the *q*_c_ according to Equation (20) without specifying this notion. Table 11 compares that of the experimental *q*_c_ (determined by X-ray spectroscopy [85,86,87,88]) with the *q*_c_ calculated by the EN method in several crystalline compounds.

As follows from Table 11, when $\bar{q}$_c_ = 1 ± 0.3, which is well in agreement with Pauling’s principle of electro-neutrality [52]: in a stable compound, the *q*_c_ must be close to ±1e. As an illustration of this principle, we experimentally list (by the X-ray method) the measured distribution of charges in the Co and Cr complexes [89]:[Co(NH_3_)_6_]^2+^  *q*_N_ = −0.62
[Cr(CN)_6_]^2−^ *q*_N_ = −0.54
*q*_H_ = + 0.36 *q*_C_ = + 0.22
*q*_Co_ = −0.49 *q*_Co_ = −0.38

Coordination charges of high-EN elements (Cu, Au, Hg, Tl, Pt) in the low valence state in some crystalline compounds, according to the EN method, must be < 0 [90,91,92,93,94,95,96]; later, ESCA studies confirmed this conclusion with the example of platinum and gold compounds [97,98,99,100,101,102,103].

Consider, as an example, two models of distribution charges in compounds of univalent gold and divalent platinum: either ${\mathrm{Au}}^{+}X^{-}$ and ${\mathrm{Pt}}^{+q}X_{2\;}^{-q}$ (classical chemistry), or ${\mathrm{Au}}^{-}X^{+}$ and ${\mathrm{Pt}}^{-q}X_{2}^{+q}$ (concept of EN). In fact, the observed oxidation reactions occur according to the schemes:AuI + Cl_2_ → AuICl_2_ and PtI_2_ + Cl_2_ → PtI_2_Cl_2_
which confirm the EN predictions. All possible mixed tetra-halides of Pt and tri-halides of Au, di-halides of Cu [95] and Hg [92], and chalcogenohalides of Tl [93] were synthesized by the oxidation of metals owing to their negative charges.

Interestingly, the mixed halides of Pt^IV^ have different properties depending on the order; in the case of halogens, these were added as follows: PtX_2_ + Y_2_ or PtY_2_ + X_2_. The structure of PtX_2_ has a motif of squares with shared vertices, and additional halogens complete these to form octahedra in PtX_2_Y_2_, while in PtY_2_X_2_, the atoms of Y are in squares, but additional atoms of X are in the vertical coordinate. Such compounds were named the square-coordinate isomers [91]. TlSeBr also showed different properties depending on the route of synthesis: TlBr + Se → Se = Tl^III^–Br or 2Tl + Se_2_Br_2_ → 2Tl^I^–Se–Br. The given structural formulae are confirmed by IR-spectroscopy and DSC study; these compounds were named the valence isomers [93].

Studying literatures showed that similar reactions were observed by other researchers in Fe, Sb, Cr, Re, W, and U compounds with a variety of ligands, such as halogens, chalcogens, SCN, N_3_, NO_3_, CO_3_, SO_4_ and methyl. Thus, an oxidation reaction of the polyvalent metals in solid compounds is the chemical method that allows for the estimation of the sign of their coordination charges.

### 3.5. Change of Chemical Bonding under Pressure

The effect of high pressure on the bond ionicity in crystalline compounds is a very interesting but insufficiently studied phenomenon. For a long time, the only source of experimental information about this effect was the study of pressure dependences upon the optical and polarization properties of crystals, and their explanation by Szigeti’s equation. These studies show that the *e** of atoms in crystals under high pressures usually decreases, but in AgI, TlI, HgTe, AlSb, GaN, InAs, PbF_2_ and SiO_2_ it increases with pressure (see references in [18]). Difficulties in measuring the optical properties of crystals under high pressures, increasing anharmonicy of vibrations, and the deformation of IR absorption bands, reduces the precision of determining the effective charges of atoms in compressed crystal. Therefore, it is desirable to find an independent method to solve this problem. Such a method can be the EN technique, see [104,105].

Let reaction M + X → MX have the thermal effect *Q* at ambient thermodynamic conditions, and the compression works of the initial reagents and the final product under pressure denoted as *W_c_*. Obviously, if *W*_c_ (mixture) > *W*_c_ (compound) > 0, then Δ*W*_c_ should be subtracted from the standard heat effect to yield the *Q* corresponding to high pressures, and vice versa. Usually Δ*W*_c_
*>* 0, hence under pressure, the *Q* and Δχ (bond ionicity) decrease. The results of such an approach qualitatively agrees with the experiment, except for alkali hydrides, where the calculations predict the *e** to fall to 0, already at several tens of GPa, while in fact, no change of electronic structure occurred up to 100 GPa and even higher [106,107]. This contradiction can be resolved by taking into account that the compression work only partly goes into changing the chemical bonds. Thus, a comparison of the *W*_c_ of mixtures and compounds allows us to define a change in *Q* (hence of the ENs of atoms and bond polarity) on variation of the pressure. The reduction of *Q* and the bond polarity under pressure is observed in crystals of AB-type, viz. Group 1—Group 17, Group 2—Group 16, Group 13—Group 15 compounds. Table 12 shows that the *∂e***/∂P* for these crystals decreases by 10^−4^ to 10^−3^ GPa^−1^. Szigeti’s method predicts the same signs and similar absolute values, *∂e*/∂P* = 3.3 × 10^−4^ GPa^−1^. Remarkably, the increase in *Q*(*P*) in CuX, AgX, and TlX under high pressures indicates an increase in polarity; the corresponding data using Szigeti’s method are not available.

Such behavior of substances under pressure is caused by the different compressibility of cations and anions. If the anion is softer than the cation (e.g., in halides of Cu, Ag and Tl), then it will be compressed more strongly and the *∂e*/∂P* > 0, i.e., the bond polarity, will increase with pressure. The electronegativity of the anion, being inversely related to the atomic size, will increase and, hence, so will Δχ, if χ_X_ > χ_M_. In the case of a softer cation (e.g., in alkali halides), the χ_M_ on compression will increase more strongly than χ_X_, and the ionicity will decrease.

## 4. Concluding Remarks

The concept of EN, like other methods of theoretical chemistry, has its critics, which can be divided into two groups—some researchers criticize the particular approach to the calculation of EN and propose new, more successful ones (from their point of view); others see fundamental defects in the concept of EN and suggest it be completely excluded from chemistry.

The first criticism came from Fajans [108], who pointed out that in the series HC ≡ CCl → H_2_C=CHCl → H_3_C–CH_2_Cl, the charge of Cl changes the sign from +1 to 0 to −1, which contradicts the notion of a constant EN of the carbon atom. In fact, χ(C) depends on the state of the hybridization of carbon, being 2.5 for sp^3^, 2.9 for sp^2^ and 3.2 for sp, and since χ(Cl) = 3.0, this explains the reversion of its charge; this was noted in the author’s letter to Fajans. Hückel [109] criticized the EN dimension (energy^½^), as physically meaningless; however, to calculate bond ionicity, Pauling used the values Δχ^2^; we noted [110] a definite analogy between Δχ and the wave function ψ, bearing in mind that only |ψ|^2^ has physical significance. Iczkowski and Margrave [111] draw attention to the different dimensions of ENs of different methods. Syrkin [112] stated that the EN concept cannot explain modern data, ignoring the agreement of the XRS atomic charges in crystals with the results of the EN method [113]. Spiridonov and Tatevsky [114] noted that the EN concept, using the atom in the molecule approach, contradicts (?) the philosophy of quantum chemistry. Comments on this irrational criticism are given in works [82,115]. A summary of the acute discussion of the concept of EN, which took place in the Soviet Union in 1962, can be found in [116].

The arguments in favor of EN can be summarized thus. The fact that EN is defined through different observed properties, and so has a non-unique dimensionality, merely reflects the multi-faceted nature of the chemical bond. Indeed, this can be an asset rather than a liability, as EN can serve as a nodal point connecting the various physical characteristics of a substance, hence its wide usage in chemistry. A certain ‘fuzziness’ of the EN concept is really typical for chemistry cf. the notions of metallicity, acidity, reactivity, etc. Sixty years have passed since the stormy discussions and condemnations of the EN concept, but its development and application in various fields of science, such as structural chemistry, materials science, molecular spectroscopy, and various fields of physical and inorganic chemistry [117,118,119,120,121] does not stop; it was even suggested to use EN as the third coordinate in the Periodic Table [122].

Pauling’s concept of EN formed the quantitative dependence of the thermodynamics of substances on the chemical bonding within them. This foundation provided the successful development and application of the EN concept in chemistry.

## Figures and Tables

**Figure 1 molecules-27-08215-f001:**
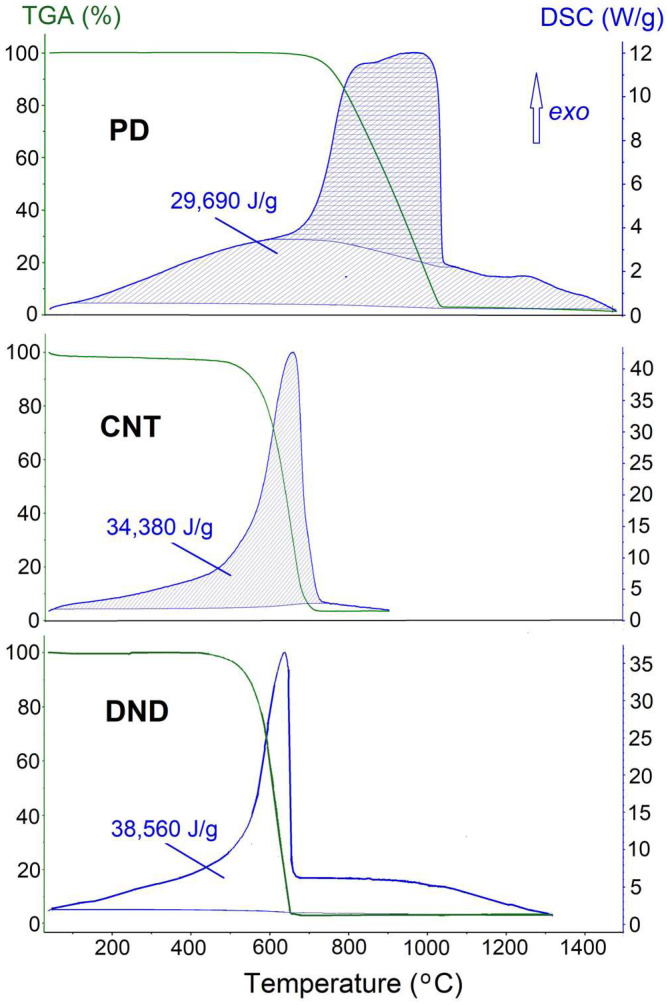
DSC/TGA curves for the oxidation of carbon with different grain sizes.

**Figure 2 molecules-27-08215-f002:**
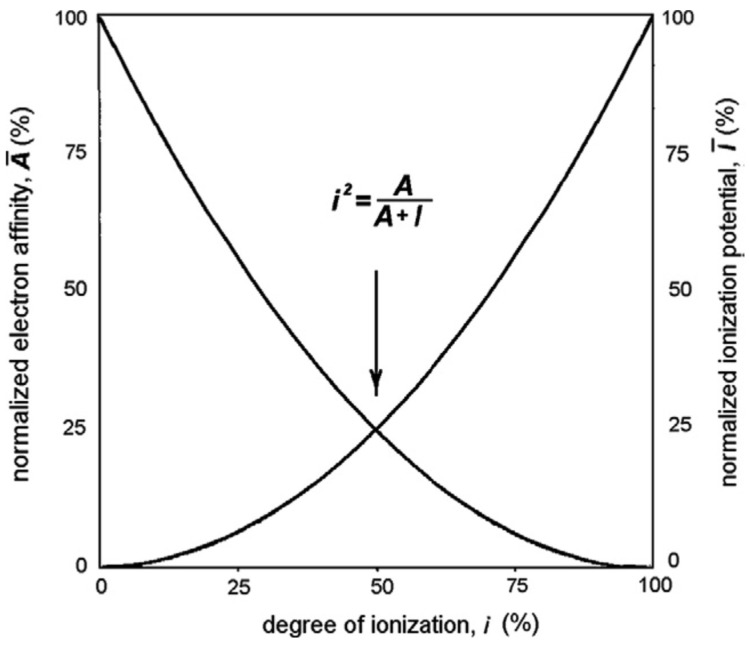
Schematic presentation of *I* and *A* as functions of *i*; the curves correspond to Equation (15).

**Table 1 molecules-27-08215-t001:** Ionization electronegativity of free atoms.

Li	Be	B	C	N	O	F			
0.90	1.45	1.90	2.37	2.85	3.32	3.78			
Na	Mg	Al	Si	P	S	Cl			
0.89	1.31	1.64	1.98	2.32	2.65	2.98			
K	Ca	Sc	Ti	V	Cr	Mn	Fe	Co	Ni
0.81	1.17	1.50	1.25 ^b^	1.60 ^c^	1.33 ^b^	1.32 ^b^	1.35 ^b^	1.38 ^b^	1.40 ^b^
			1.86	1.92 ^d^	1.66 ^c^	1.70 ^c^	1.67 ^c^	1.72 ^c^	1.76 ^c^
				2.22	1.98 ^d^	2.03 ^d^			
Cu	Zn	Ga	Ge	As	Se	Br			
1.48	1.64	1.83	2.09	1.70 ^c^	2.61	2.88			
1.66 ^b^				2.27					
Rb	Sr	Y	Zr	Nb	Mo	Tc	Ru	Rh	Pd
0.80	1.13	1.41	1.23 ^b^	1.53 ^c^	1.92 ^d^	1.88 ^d^	1.35 ^b^	1.39 ^b^	1. 45 ^b^
			1.71	2.02	2.33		1.93 ^d^	1.94 ^d^	2.01 ^d^
Ag	Cd	In	Sn	Sb	Te	I			
1.57	1.65	1.80	1.29 ^b^	1.60 ^c^	2.45	2.68			
			2.00	2.24					
Cs	Ba	La	Hf	Ta	W	Re	Os	Ir	Pt
0.77	1.08	1.35	1.28 ^b^	1.54 ^c^	1.83 ^d^	1.85 ^d^	1.39 ^b^	1.40 ^b^	1.45 ^b^
			1.71	1.99	2.28	2.52	1.86 ^d^	1.89 ^d^	1.95 ^d^
Au	Hg	Tl	Pb	Bi	Po	Th	U		
1.78	1.79	0.96 ^a^	1.31 ^b^	1.58 ^c^	2.29	1.59 ^d^	1.68 ^d^		
1.93 ^c^		1.89	2.07	2.13					

valence ^a^ *v* = 1, ^b^ *v* = 2, ^c^ *v* = 3, ^d^ *v* = 4.

**Table 2 molecules-27-08215-t002:** Thermochemical electronegativity of metals in molecular halides.

M (I)	χ	M (II)	χ	M (II)	χ	M (III)	χ	M (IV)	χ
Li	0.98	Cu	2.04	Hg	1.95	Sc	1.34	Ti	1.77
Na	0.93	Be	1.43	Sn	1.42	Y	1.25	Zr	1.63
K	0.72	Mg	1.38	Pb	1.55	La	1.12	Hf	1.56
Rb	0.67	Ca	1.00	Cr	1.53	B	1.96	C	2.60
Cs	0.53	Sr	0.95	Mn	1.35	Al	1.64	Si	2.05
Cu	1.46	Ba	0.82	Fe	1.58	In	1.73	Ge	2.15
Ag	1.61	Zn	1.60	Co	1.64	Ga	1.73	Sn	2.06
Au	1.84	Cd	1.70	Ni	1.71	V	1.80	Pb	2.17
Tl	1.06					Cr	1.92	W	2.19

**Table 3 molecules-27-08215-t003:** Ionization electronegativity ($\chi_{s}$) of atoms in solid metals.

Li	Be	B	C						
0.47	0.65	1.17	1.29						
Na	Mg	Al	Si						
0.41	0.62	1.03	1.18						
K	Ca	Sc	Ti	V	Cr	Mn	Fe	Co	Ni
0.38	0. 52	0.70	0.98	1.03	1.03	0.89	0.98	0.97	1.16
Cu	Zn	Ga	Ge	As	Se				
0.93	0.64	1.14	1.28	1.18	1.46				
Rb	Sr	Y	Zr	Nb	Mo	Tc	Ru	Rh	Pd
0.35	0.50	0.65	0.97	1.05	1.17	1.01	1.32	1.34	1.30
Ag	Cd	In	Sn	Sb	Te	I			
1.02	0.76	1.11	1.01	0.98	1.34	1.78			
Cs	Ba	La	Hf	Ta	W	Re	Os	Ir	Pt
0.28	0.46	0.66	0.80	0.83	1.14	1.22	1.20	1.28	1.32
Au	Hg	Tl	Pb	Bi	Th	U			
1.23	0.82	0.78	1.07	1.14	0.89	0.97			

**Table 4 molecules-27-08215-t004:** Thermochemical electronegativity ($\chi_{s}$) of metals in solid compounds.

M (I)	$\chi_{s}$	M (II)	$\chi_{s}$	M (II)	$\chi_{s}$	M (III)	$\chi_{s}$	M (IV)	$\chi_{s}$
Li	0.55	Cu	1.33	Sn	1.09	Sc	0.69	Ti	0.68
Na	0.48	Be	0.98	Pb	1.04	Y	0.72	Zr	0.77
K	0.37	Mg	0.68	Cr	0.96	La	0.58	Hf	0.75
Rb	0.35	Ca	0.44	Mn	0.73	Al	1.00	C	2.60
Cs	0.32	Sr	0.41	Fe	1.18	In	1.73	Si	2.05
Cu	1.25	Ba	0.42	Co	1.07	Ga	1.19	Ge	2.15
Ag	1.30	Zn	0.85	Ni	1.24	V	0.88	Sn	0.81
Au	2.01	Cd	0.91	Pd	1.55	Nb	0.66	Pb	1.53
Tl	0.85	Hg	1.44	Pt	1.64	Sb	1.27	Th	0.82
						Bi	1.22	U	0.95
						Cr	1.01	W	1.14

**Table 5 molecules-27-08215-t005:** Bulk and average coordination numbers of particles, cohesive energies (kJ/mol) in bulk and nanosized cubic elements.

M	*N_b_*	$\bar{N}$	*E_b_*	*E_p_*	M	*N_b_*	$\bar{N}$	*E_b_*	*E_p_*
Ac	12	9.828	406	332	Mn	12	10.42	283	246
Ag	12	10.33	285	245	Mo	8	6.863	659	565
Al	12	10.34	331	285	Na	8	6.450	107	86.7
Au	12	10.33	368	317	Nb	8	6.849	733	627
Ba	8	6.186	179	138	Ni	12	10.56	430	378
C	4	3.629	717	650	Pb	8	9.976	195	162
Ca	12	9.717	178	144	Pd	12	10.41	377	327
Co	12	10.55	427	375	Pt	12	10.40	566	490
Cr	8	6.958	397	346	Ra	8	6.140	159	122
Cs	8	5.781	76.5	55.3	Rb	8	5.939	80.9	60.0
Cu	12	10.69	337	301	Rh	12	10.44	556	434
Fe	8	6.964	415	362	Si	4	3.435	450	386
Ge	4	3.412	372	317	Sr	12	6.205	164	127
Ir	12	10.43	609	529	Ta	8	6.807	782	671
K	8	6.077	89.0	67.6	Th	12	9.921	602	498
La	12	9.839	431	353	V	8	6.907	515	445
Li	8	6.732	159	134	W	8	6.856	851	729

**Table 6 molecules-27-08215-t006:** Electronegativities of nanosized elements.

M	*N_s_*	*E_s_*	*E_b_*/*I_o_*	*I_s_*	χ_nano_	M	*N_s_*	*E_s_*	*E_b_*/*I_o_*	*I_s_*	χ_nano_
Ac	7.181	243	0.3652	6.896	1.02	Mn^II^	7.181	169	0.2545	6.902	1.02
Ag	7.181	170	0.3897	4.534	0.83	Mo^IV^	4.523	372	0.3012	12.82	1.40
Al	7.181	198	0.1931	10.63	1.27	Na	4.523	60. 8	0.2168	2.905	0.66
Au	7.181	220	0.4136	5.520	0.92	Nb^III^	4.523	414	0.4941	8.692	1.15
Ba	4.523	101	0.2440	4.298	0.81	Ni^III^	7.181	257	0.2192	112.2	1.36
C	2	358	0.2007	18.50	1.68	Pb^II^	4.523	110	0.1802	6.349	0.98
Ca	7.181	106	0.2049	5.382	0.90	Pd^II^	7.181	225	0.2811	8.310	1.12
Co^III^	7.181	255	0.2269	11.66	1.33	Pt^II^	7.181	338	0.4261	8.233	1.12
Cr^III^	4.523	225	0.2280	10.21	1.25	Ra	4.523	89.9	0.2136	4.361	0.81
Cs	4.523	43.2	0.2036	2.201	0.58	Rb	4.523	45.7	0.2007	2.362	0.60
Cu	7.181	202	0.4526	4.623	0.84	Rh^III^	7.181	333	0.3054	11.29	1.31
Fe^III^	4.523	235	0.2359	10.32	1.25	Si	2	225	0.1809	12.89	1.40
Ge	2	186	0.1586	12.15	1.36	Sr	7.181	98.1	0.2032	5.003	0.87
Ir^III^	7.181	364	0.3510	10.76	1.28	Ta^III^	4.523	442	0.5192	8.825	1.16
K	4.523	50.3	0.2125	2.454	0.61	Th^IV^	7.181	360	0.3818	9.777	1.22
La	7.181	258	0.3729	7.168	1.04	V^III^	4.523	291	0.3162	9.551	1.20
Li	4.523	90.1	0.3062	3.048	0.68	W^IV^	4.523	481	0.4864	10.25	1.25

**Table 7 molecules-27-08215-t007:** Dipole moments (in *D*) and bond polarities in MX molecules.

M	F		Cl		Br		I	
	*μ*	*p*	*μ*	*p*	*μ*	*p*	*μ*	*p*
H	1.826	0.41	1.108	0.18	0.827	0.12	0.448	0.06
Li	6.327	0.84	7.129	0.73	7.226	0.69	7.428	0.65
Na	8.156	0.88	9.002	0.79	9.118	0.76	9.236	0.71
K	8.592	0.82	10.27	0.80	10.63	0.78	10.82	0.74
Rb	8.546	0.78	10.51	0.78	10.86	0.77	11.48	0.75
Cs	7.884	0.70	10.39	0.73	10.82	0.73	11.69	0.73
Cu	5.77	0.69	5.2 ^a^	0.53				
Ag	6.22	0.65	6.08	0.55	5.62	0.49	4.55	0.37
Au	4.32	0.47	3.69 ^b^	0.35				
Tl	4.228	0.42	4.543	0.38	4.49	0.36	4.61	0.34

^a^ [53], ^b^ [54].

**Table 8 molecules-27-08215-t008:** IRS-bond ionicity (e***/***v*) in molecules.

Molecule	*i*	Molecule	*i*	Molecule	*i*	Molecule	*i*	Molecule	*i*
HF	0.382	OH_2_	0.236	CH_4_	0.028	BF_3_	0.506	LiF	0.861
HCl	0.184	NH_3_	0.034	SiH_4_	0.226	BCl_3_	0.249	LiCl	0.760
HBr	0.114	NF_3_	0.385	GeH_4_	0.216	CF_4_	0.512	NaF	0.889
HI	0.040	PH_3_	0.119	SnH_4_	0.254	CCl_4_	0.261	NaCl	0.809
LiH	0.654	PF_3_	0.580			CO_2_	0.268	KCl	0.830

**Table 9 molecules-27-08215-t009:** Bond ionicity in MX halides.

MX	*i* _B_	*i* _P_	*i* _S_	MX	*i* _B_	*i* _P_	*i* _S_
LiF	0.67	0.78	0.75	CuF	0.61	0.66	0.37
LiCl	0.63	0.55	0.67	CuCl	0.56	0.38	0.28
LiBr	0.62	0.48	0.62	CuBr	0.55	0.31	0.24
LiI	0.60	0.38	0.54	CuI	0.53	0.21	0.15
NaF	0.68	0.80	0.80	AgF	0.61	0.61	0.41
NaCl	0.64	0.57	0.71	AgCl	0.57	0.33	0.33
NaBr	0.63	0.50	0.67	AgBr	0.55	0.26	0.28
NaI	0.61	0.40	0.58	AgI	0.54	0.16	0.20
KF	0.71	0.84	0.85	AuF	0.57	0.53	
KCl	0.67	0.64	0.76	AuCl	0.53	0.25	
KBr	0.66	0.57	0.72	AuBr	0.52	0.18	
KI	0.64	0.47	0.64	AuI	0.50	0.13	
CsF	0.73	0.87	0.97	TlF	0.65	0.76	0.64
CsCl	0.69	0.69	0.89	TlCl	0.61	0.53	0.55
CsBr	0.68	0.64	0.84	TlBr	0.59	0.46	0.51
CsI	0.66	0.54	0.76	TlI	0.56	0.35	0.43

**Table 10 molecules-27-08215-t010:** Effective charges of atoms in MX crystals.

M	F		Cl		Br		I	
	*i* _s_	*e**	*i* _s_	*e**	*i* _s_	*e**	*i* _s_	*e**
Li	0.81	0.81	0.78	0.77	0.76	0.74	0.73	0.54
Na	0.81	0.83	0.78	0.78	0.76	0.75	0.74	0.74
K	0.82	0.92	0.79	0.81	0.77	0.77	0.75	0.75
Rb	0.82	0.97	0.80	0.84	0.77	0.80	0.75	0.77
Cs	0.84	0.96	0.82	0.85	0.80	0.82	0.78	0.78
Cu	0.71		0.68	0.98	0.65	0.96	0.62	0.91
Ag	0.71	0.89	0.68	0.71	0.65	0.67	0.57	0.61
Tl	0.74		0.71	0.88	0.68	0.84	0.65	0.83

**Table 11 molecules-27-08215-t011:** Coordination effective charges of metals in compounds.

Metal	Compounds	*q*_c_ (exp)	*q*_c_ (cal)	Metal	Compounds	*q*_c_ (exp)	*q*_c_ (cal)
Cr	CrSO_4_·7H_2_O	1.9	1.8	Co	Co(NO_3_)_3_	1.2	0.6
	Cr(NO_3_)_3_	1.2	1.3		Co(C_5_H_5_)_2_	0.4	0.7
	K_2_CrO_4_	0.1	0.5		Co(C_5_H_5_)_2_Cl	1.0	0.9
	Cr(C_6_H_6_)_2_	1.3	1.4	Ni	Ni(C_5_H_5_)_2_	0.7	0.6
Mn	Mn(NO_3_)_2_·4H_2_O	1.8	1.8		Ni(C_5_H_5_)_2_Cl	1.0	0.8
	K_3_Mn(CN)_6_	0.9	0.6	Os	OsO_2_	0.8	0.7
	Mn(C_5_H_5_)_2_	1.5	1.3		K_2_OsCl_6_	0.8	0.5
Fe	(NH_4_)_2_Fe(SO_4_)_2_·6H_2_O	1.9	1.7		K_2_OsO_4_	0.8	0.7
	K_3_Fe(CN)_6_	1.0	0.4		K_2_OsNCl_5_	0.7	0.8
	Fe(C_5_H_5_)_2_	0.6	0.7		KOsO_3_N	1.0	0.9

**Table 12 molecules-27-08215-t012:** Change of atomic charges (−*∂e*/∂P*, 10^−4^ GPa^−1^) in crystals MX under *P* = 10 GPa.

M^I^	F	Cl	Br	I	M^II^	O	S	Se	Te
Li	5.2	1.5	0	1.9	Be	0	−1.6	1.6	3.7
Na	7.0	2.4	1.5	4.0	Mg	0.9	0	0	5.6
K	11.7	5.9	5.4	4.1	Ca	3.7	2.2	3.3	7.5
Rb	10.6	3.9	2.4	1.7	Sr	7.0	5.9	7.2	9.6
Cs	10.4	6.7	8.3	7.6	Ba	16.5	16.5	18.0	24.6
Cu		−8.4	−4.7	6.9	Zn	0	−2.2	0	4.4
Ag	−3.7	−5.6	−3.4	−6.6	Cd	0	−2.8	0	5.9
Tl	−12.6	−12.5	−11.6	−10.9	Hg	−5.9	−2.8	0.6	10.0
**M^III^**	**N**	**P**	**As**	**Sb**	Sn	−12.2	−2.8	0	0
B	0.6	9.7	4.7		Pb	−7.8	−3.7	0	4.7
Al	0.6	5.3	0.6	0	Mn	−0.4	−4.0		1.6
Ga	2.2	9.4	2.5	2.8	
In	6.9	10.0	1.9	1.9
La		15.3	14.7	14.4
Th	2.5	7.5		
U	0	3.4

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
