# Peer review of "Energy Electronegativity and Chemical Bonding"

_molecules, 2022, doi:10.3390/molecules27238215_

Round 1

Reviewer 1 Report

Energy electronegativity and chemical bonding by Batsanov.

In this manuscript, the author describes several approaches to obtaining the electronegativity of an atom in different environments. The manuscript is interesting, but it seems more like a review because it is difficult to identify where the author's contribution starts in each section. The authors should clarify where the new contribution begins and briefly describe the paper sections at the beginning of the manuscript.  Also, the author used shallowly described concepts, for example, the concepts from QTAIM. Some acronyms are not defined as  E_MX^cov in eq. 5. Eq. 9 is missing. Figure 1 is duplicated. 

Author Response

In this manuscript, the author describes several approaches to obtaining the electronegativity of an atom in different environments. The manuscript is interesting, but it seems more like a review because it is difficult to identify where the author's contribution starts in each section. The authors should clarify where the new contribution begins and briefly describe the paper sections at the beginning of the manuscript.

Yes, the paper consists of three parts: a general historical review, a summary of my own publications on EN, and the new, previously unpublished results. The description, as recommended by the referee, is added at the end of the Introduction (Section 1). Also, the Abstract is re-written to make a clear distinction between the review parts and new work.

Also, the author used shallowly described concepts, for example, the concepts from QTAIM.

Actually, QTAIM is only peripherally related to the main theme of the paper, as one of many approaches to estimating effective charges of atoms. 

Some acronyms are not defined as E_MX^cov in eq. 5.

The explanation is added after Eq. 5.

Eq. 9 is missing.

The numbering has been corrected.

Figure 1 is duplicated.

I apologize for the mistake. The figure was supposed to show the diagrams for synthetic diamond powder (PD), nanotubes (CNT) and detonation nanodiamond (DND). The PD plate was repeated twice, while CNT was missing. The correct figure is now provided, as a single plate to avoid any mishap.

Reviewer 2 Report

In this review article , author has nicely correlated  the electronegativity of atoms, ions, molecules and nano structures step by step. The manuscript is well written and organised.  Although it is quite important to know the relationship between electronegativity and chemical bonding for different aforementioned cases ,still many thing are not clear in the present version of the manuscript. I suggest before accepting, Author should address the following comments.

(1) For isolated atoms, ions even for molecules, the theory of electronegativity is well known. Author should focus more on the nano structure part.

(2) How the DSC and TGA data can help to understand the chemical bonding of a carbon nanostructure is not well explained.

(3) Author have mentioned in the abstract regarding the nitrogen fixation with nano diamonds. But later in the main text or conclusion there are no discussions. I think this should be modified.

(4) If author can write few more lines regarding the relationship between dangling bonds and electronegativity of nano structures where surface plays an important role.

Author Response

In this review article, author has nicely correlated the electronegativity of atoms, ions, molecules and nano structures step by step. The manuscript is well written and organised.  Although it is quite important to know the relationship between electronegativity and chemical bonding for different aforementioned cases, still many things are not clear in the present version of the manuscript. I suggest before accepting, Author should address the following comments.

(1) For isolated atoms, ions even for molecules, the theory of electronegativity is well known. Author should focus more on the nano structure part.

The nanomaterials part has been presented in more detail. However, I believe that in more traditional fields, the development of the EN concept has not yet run its course: witness the very recent work of Oganov et al., whose modification of Pauling’s original (1932) approach suddenly revitalized the apparently long-closed chapter.

(2) How the DSC and TGA data can help to understand the chemical bonding of a carbon nanostructure is not well explained.

Essentially, it shows that the cohesive energy of carbon particles decreases together with size; the comment added to Section 2.4 (page 7).

(3) Author have mentioned in the abstract regarding the nitrogen fixation with nano diamonds. But later in the main text or conclusion there are no discussions. I think this should be modified.

The description was added to Section 2.4 (penultimate paragraph).

(4) If author can write few more lines regarding the relationship between dangling bonds and electronegativity of nano structures where surface plays an important role.

The discussion is added to Section 2.4 (second paragraph).

Round 2

Reviewer 1 Report

The author has addressed all my comments for this paper. In this version, it is possible to detect the author's contributions. The author fixed all the issues noted in the previous version.